# Symbolic Regression in Financial Economics

**Jiacheng Liu**
Purdue University
`jiacheng-liu@purdue.edu`

**Siqi Guo**
Purdue University
`guo477@purdue.edu`

## Abstract

We apply symbolic regression, the machine learning approach of recovering models from data, in financial economics. Specifically, we present a data set consisting of equations that cover a broad range of topics in financial economics. These equations are built off a common set of mathematical symbols but importantly have new variations in functional forms. We test the joint performance of deep learning and genetic programming symbolic regression systems in recovering these non-physical equations.

## 1 Introduction

Symbolic regression (SR) is a branch of machine learning that aims to recover mathematical expressions from data. SR is special in its strong interpretability: the recovered equations allow for direct human inspection (de Franca et al., 2023). SR can also assist scientific discovery in finding accurate yet relatively concise fundamental equations.

Recent progress in symbolic regression has been promising. Although SR has been proven to be an NP-hard problem in general (Virgolin & Pissis, 2022), literature has proposed several viable approaches, including Genetic Programming (e.g., Schmidt & Lipson (2009)) and Deep Learning (e.g., Kamienny et al. (2022)). Current SofA methods achieve high recovery rates for physical equations. For example, AI Feynman (Udrescu & Tegmark, 2019) recovered all 100 equations from the Feynman Lectures on Physics. GSR (Tohme et al., 2022) further achieved perfect recovery rates in a range of datasets. On the other hand, the performance of symbolic regression methods on non-physical equations is still largely unknown. This is at least partly due to the lack of a benchmark dataset for non-physical systems. In this work, we present a ground-truth database based on financial economics that complements existing physical systems benchmarks.

Our main contribution is extending the scope of SR from traditionally-focused physical equations into financial economics. In particular, the financial system is one of the most complex yet mature human-made systems. A closely related contribution is to provide a preliminary out-of-sample test for existing SR methods.

## 2 Related Work

Our paper relates to two strands of literature. First, our paper is related to the papers providing SR benchmarks (Udrescu & Tegmark (2019), Cava et al. (2021), Romano et al. (2020)). Our paper differs and complements this line of prior work by focusing on the set of equations that comes from financial systems and not the physical world.

Second, we connect with the literature on applications of SR in social sciences. Duffy & Engle-Warnick (2002) applied SR to repeated ultimatum games. Their algorithm uncovers relatively simple strategies while encountering difficulties with more random strategies. Yang et al. (2015) model oil productions and Pan et al. (2019) model carbon emissions in OECD countries. Truscott & Korns (2014) use SR in predicting unemployment rates. Balla et al. (2022) demonstrate OccamNet is applicable to a wide range of social science topics including network structure, epidemic spread, economic production, and growth. Our work differs both in methods and in our focus on systematic equations from financial economics.

## 3 METHODS

Inspired by AI Feynman (Udrescu & Tegmark, 2019), we select 14 equations that are suitable for SR from the appendix of the seminar textbook by Ross et al. (2021) and generate synthetic data. We select non-linear equations that are economically meaningful and foundational, at the same time, have moderate complexity. See Appendix B for more context of these equations. We exclude equations that are either trivial (e.g., simple linear equations with few inputs) or equations that are too challenging or out-of-distribution (e.g., differential equations). Our choice of equations is conservative, in the sense that the global recovery rate on all equations is likely higher due to a larger number of trivial equations excluded. For each equation tested, we randomly sample all inputs independently from [0,1) interval, then we fit them into the given mathematical expression to obtain the target. This operation is repeated for 1000 draws for each equation.

## 4 EXPERIMENTS

We evaluate the performance of two representative SR methods in recovering equations from data. For genetic programming (GP) -based models, we select the commercial software Eureqa Schmidt & Lipson (2009) and allow up to 300 seconds for Eureqa to solve the equation. The program has access to constants in addition to a set of mathematical operations $(+, -, \times, \div, \hat{}, \sqrt{.}, \exp)$. For deep learning (DL)-based models, we select the transformer-based end-to-end model (Kamienny et al., 2022). We tested the pre-trained transformer model with 3, 7, or 20 beams. We use the pre-trained model and therefore defer readers to the original paper for more training details. If the exact or mathematical equivalent expressions were recovered by either method, we consider an equation solved.

Table 1 reports the results from experiments. We report that both DL and GP are capable of recovering a considerable set of nonlinear equations in our newly assembled dataset. Both methods combined solved 9 out of 14 equations. We view our results as preliminary and lower bounds of the true potential of these methods. We refrain from making conclusions about the relative performance of DL and GP methods. Overall, our work implies that SR is capable of recovering many equations of the financial system.

The dataset is available at https://github.com/Jiacheng-Liu/Symbolic-Regression-in-Financial-Economics.

Table 1: Tested equations: the first two columns report equation index and mathematical expression. Deep learning (DL) and genetic programming (GP) test results are reported.

| Eq. Index | Equation Expression | Solved by DL | Solved by GP |
|---|---|---|---|
| 7.1 | $P = C\frac{1-\frac{1}{(1+r)^t}}{r} + \frac{F}{(1+r)^t}$ | No | No |
| 8.1 | $P_0 = \frac{D+P_1}{1+R}$ | Yes | Yes |
| 8.4 | $P_t = \frac{D_{t+1}}{R-g}$ | No | **Yes** |
| 8.5 | $P_0 = \frac{D_1}{R-g_1}[1-(\frac{1+g_1}{1+R})^t] + \frac{P_t}{(1+R)^t}$ | No | No |
| 14.6 | $WACC = (\frac{E}{V})R_E + (\frac{D}{V})R_D(1-T_c)$ | No | **Yes** |
| 16.1 | $R_E = R_A + (R_A - R_D)\frac{D}{E}$ | Yes | Yes |
| 19A.4 | $C^* = \sqrt{\frac{2TF}{R}}$ | No | **Yes** |
| 19A.5 | $C^* = L + (\frac{3}{4}F\frac{\sigma}{R})^{\frac{1}{3}}$ | No | No |
| 20.4 | $PV = \frac{(P-v)(Q'-Q)}{R}$ | Yes | Yes |
| 20.14 | $Q^* = \sqrt{\frac{2TF}{CC}}$ | No | **Yes** |
| 21.3 | $E(S_t) = S_0[1 + (h_{FC} - h_{US})]^t$ | No | No |
| 21.4 | $F_1 = S_0\frac{1+R_{FC}}{1+R_{US}}$ | **Yes** | No |
| 21.6 | $F_1 = S_0[1 + (R_{FC} - R_{US}]$ | Yes | Yes |
| 25.8 | $V = E \times e^{-Rt} - P$ | No | No |

URM STATEMENT

The authors acknowledge that all key authors of this work meet the URM criteria of ICLR 2023 Tiny Papers Track.

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

## A  OVERVIEW OF SYMBOLIC REGRESSION

Symbolic regression (SR) is a type of machine learning that aims to discover mathematical expressions that accurately describe a given dataset. SR is more general than traditional regressions because it allows the search for both symbolic models and parameters that best fit the data. In particular, the final model is represented as a human-readable equation or a combination of mathematical operations and functions.

More formally, the task of symbolic regression is to automatically derive a mathematical expression $f(.) \in F$ that predicts the target variable $y \in R$ based on the input variables $X \in R^d$, where $F$ is the space of all possible functional forms formed by mathematical operations. SR aims to find a concise and interpretable mathematical representation of the underlying relationship in the data.

More specifically, in this work, we perform our analysis in two steps. We first take equations as given and generate data points accordingly. We then test if either SR model can recover the exact equations from the generated data. An equation is considered solved if the expressions recovered are mathematically equivalent. One limitation of our work is we did not evaluate $R^2$. Therefore, even if the exact expression was not recovered, it is still possible that a model solution is a relatively good approximation.

## B  SCOPE OF EQUATIONS TESTED

The equations tested are economically meaningful equations from different fields of financial economics. A brief breakdown is as follows: Equations 7.1 to 8.5 are asset pricing equations, namely the target is to provide an estimate of price given the cash flow of a bond or stock. Equation 14.6 provide the cost of capital given the capital structure and required returns as well as the tax rate. Equation 16.1 is the M&M Proposition II which calculates the cost of equity.

Equations 19A.4 and 19A.5 are used in cash and liquidity management to determine the optimal initial cash balance and cash balance target respectively. Equation 20.4 is for calculating the present value of the future incremental cash flow. Equation 20.14 determines the economic order quantity that minimizes the total inventory cost.

Equations 21.3 to 21.6 are from international finance and describe the dynamics of foreign exchange rates. We leave some equations untested, including, for example, various accounting identities and option pricing formulas. This choice has less to do with the economic significance but more to do with the focus on putting SR methods to the test. In this regard, equations should not be trivial, namely should involve more than a simple linear relationship with few inputs. At the same time, we also leave high-complexity equations for full-scale future work.

