# OpenReview forum: "Symbolic Regression in Financial Economics "
_ICLR.cc/2023/TinyPapers — Submitted to Tiny Papers @ ICLR 2023_

### Official Review · Reviewer_ppbQ · 2023-03-26

**Confidence:** 4

**Summary Of Contributions:**

This work presents a novel application of symbolic regressions in non-physical systems, specifically in financial economics. The authors provide a ground-truth database that complements existing benchmarks and extend the scope of symbolic regression from physical equations into other domains. The joint performance of deep learning and genetic programming symbolic regression systems is tested in recovering these non-physical equations, with promising results.

**Rating:**

High Potential (HP): a submission which meets the reviewing criteria and has potential to make an impact on the field

**Strengths And Weaknesses:**

Strengths:
1. The authors provide a novel application of symbolic regressions in non-physical systems, which extends the scope of symbolic regression from physical equations into other domains.
2. The authors provide a ground-truth database that complements existing benchmarks and includes unique variations that algorithms have not seen before.

Weaknesses:
1. The paper only tests the joint performance of deep learning and genetic programming symbolic regression systems in recovering non-physical equations in financial economics, so the generalizability of the results to other non-physical systems is unclear.
2. The paper does not compare the performance of symbolic regression methods with other machine learning approaches for modeling non-physical systems, which limits the assessment of the effectiveness of symbolic regressions in this context.

**Suggested Changes:**

1. The paper could benefit from a more detailed discussion of the limitations and assumptions of symbolic regression methods in modeling non-physical systems. This would help readers better understand the scope and applicability of the proposed approach.
2. The paper could also include a comparison of the performance of symbolic regression methods with other machine learning approaches for modeling non-physical systems, such as neural networks or decision trees. This would provide a more comprehensive assessment of the effectiveness of symbolic regressions in this context and help identify areas for future research.

---

### Official Review · Reviewer_AFGo · 2023-03-30

**Confidence:** 3

**Summary Of Contributions:**

The authors present a new dataset in the field of recovering models from data and symbolic regression (SR). Their main contribution is that they extend SR with equations describing financial systems except for the most dominant physical ones that have already been studied.

**Rating:**

Great Start (GS): a submission which meets some of the reviewing criteria but has room for improvement

**Strengths And Weaknesses:**

The authors select 14 new equations, suitable for SR, that are non-linear and meaningful for financial markets. They evaluate the performance of the equations using two representative SR methods, namely genetic programming and transformer-based neural network. They mention the capability of both models in recovering each non-linear equation. The methods combined recover 9 out of 14 equations.

Despite the interesting direction, the authors propose in the field of SR and the proposed experiments, the paper lacks important details in terms of reproducibility (choice of equations, hyperparameters, data generation and training, evaluation metrics). Furthermore, regarding the discussion of the results, the analysis and the importance of their findings are rather limited.


**Suggested Changes:**

Important questions to be answered:
Why SR models can recover some of the proposed equations but struggle in some of the tasks?
Which equations could achieve stronger performance in both physical and non-physical systems?

---

### Official Review · Reviewer_dfc8 · 2023-03-31

**Confidence:** 4

**Summary Of Contributions:**

The authors present symbolic regression dataset to be used for recovering a set of financial equations.

**Rating:**

Great Start (GS): a submission which meets some of the reviewing criteria but has room for improvement

**Strengths And Weaknesses:**

Strengths
- The application of symbolic regress to financial systems is interesting and the paper addresses an important need to develop datasets in this space.


Challenges
- The methods section requires more details and explanation. It's unclear where the data is coming from. Is it synthetic data? Or is it data from the market activity? The selection criteria for the equations also seem arbitrary. It would be helpful a clear definition of "suitability" for SR.
- The use of proprietary software for the genetic programming experiments makes it difficult to reproduce the results claimed by the paper.
- The paper underspecifies the details of the transformer models used for the deep learning experiments.
- The paper has a considerable amount of typos and grammatical mistakes.


**Suggested Changes:**

The authors should consider
- adding more details in the methods section concerning the challenges described above.
- there are several typos and grammatical mistakes which should be edited
- using publicly available tools for genetic programming

---

### Comment · Area_Chair_zrdV · 2023-06-06
**Final meta-review: Invite to archive**

This work meets the threshold for archival, contents the URM statement and is deanonymized

---

### Meta-Review · Area_Chair_zrdV · 2023-04-05

**Recommendation:** Invite to archive
**Confidence:** 4

**Metareview:**

Thank you for writing this interesting paper! In this work the authors make two contributions: (i) a dataset of samples from 14 important and non-linear equations in financial economics and (ii) some preliminary results of genetic programming and deep learning based symbolic regression techniques to predict these equations from the data. The paper is clear and mostly well written although the impact is perhaps limited since only sparse details of the regression techniques are given and the dataset is small and not fully justified/motivated.

Pros:
* A simple and clearly explained dataset is provided
* Some promising results are shown proving this dataset is solvable but non-trivial.
* Mostly clear text and well explained (except some typos and comments listed below).

Cons:
* It is not clear from the text how this work differs from Balla et al. (2022)
* The title is over reaching: "non-physical systems" should be replaced with "financial economics" so as not to mislead readers.
* Potentially of limited impact: if the authors are just providing the dataset then anyone could, in theory, fairly quickly make this themselves.
   * It would be more impactful if the authors discussed **why** these equations were selected
   * It would be more impactful if the authors could discuss **how** this help the symbolic regression movement above and beyond the more typical case of studying physical systems.
* Contains some typos

The reviewers are split: two rank this as a great start (GS), another described it more favourably as having high potential (HP). Given the suggested changes are mostly minor but that the impact of this work is likely limited I'm deciding to go down the middle and recommend this paper for archive. This is also based on the fact that, in its current form, the paper is correct and clear (if not reproducible). Many of the criticisms from the reviewers are regarding the details and assumptions of the ML solutions given however I view this as the lesser of the two contribution of the paper (the other being the dataset itself, which the reviewers have less of an issue with). The authors should make the changes suggested by myself and the reviewers. For a higher recommendation the authors would need to:

* Fully and convincingly justify why these equations are the ones chosen.
* Give details of the DL and GP algorithms (hyperparams etc.) chosen, thus making the paper reproducible.
* Consider sharing the code for solving these equations. If the GP code is closed-source then at least provide the DL code.
* As suggested by reviewer ppbQ: "The paper could benefit from a more detailed discussion of the limitations and assumptions of symbolic regression methods in modeling non-physical systems. This would help readers better understand the scope and applicability of the proposed approach." since I cannot tell why this is significantly different from just studying physical systems.



**Summary:**

In this work the authors make two contributions: (i) a dataset of samples from 14 important and non-linear equations in financial economics and (ii) some preliminary results of genetic programming and deep learning based symbolic regression techniques to predict these equations from the data. The paper is clear and mostly well written although the impact is perhaps limited since only sparse details of the regression techniques are given and the dataset is small and not fully justified/motivated.

**Comments And Feedback To The Authors:**

* You must cite Ross, Westerfield and Jordan
* The acronym SR should probably be defined (even though its contextually obvious)
* "...human-made systems **including** financial economics" is incorrect since you are **only** contributing equations for financial economics.
* Please provide a caption for the table.
* Typos:
   * "...listed in 1$^1$" --> "...listed in **table** 1$^1$"
   * "symbolic regressions...in non-physical systems" --> "symbolic **regression**...**to** non-physical systems"
   * "...expression underlies a given..." --> "...expression **underlying** a given..."
   * "...unveil the players' ..." --> "...unveil the **player's** ..."
* An appendix with a little more detail explain -- even at a high level -- symbolic regression would help. For example, what does it mean for an equation to be "solved", do the algorithms have access to the constants in the equations. What is/are the dependent variable/s. Intuitively why is DL worse than GP.

**Reason For Not Giving A Higher Recommendation:**

Its impact is potentially limited for the reasons listed above and by the reviewers.

**Reason For Not Giving A Lower Recommendation:**

It's clear and understandable and makes an important contribution.

---

### Decision · Program_Chairs · 2023-04-09

Invite to archive